# Coronavirus Disease 2019 (COVID-19) Pandemic across Africa: Current Status of Vaccinations and Implications for the Future

**DOI:** 10.3390/vaccines10091553

**Published:** 2022-09-17

**Authors:** Olayinka O. Ogunleye, Brian Godman, Joseph O. Fadare, Steward Mudenda, Adekunle O. Adeoti, Adesola F. Yinka-Ogunleye, Sunday O. Ogundele, Modupe R. Oyawole, Marione Schönfeldt, Wafaa M. Rashed, Ahmad M. Galal, Nyasha Masuka, Trust Zaranyika, Aubrey C. Kalungia, Oliver O. Malande, Dan Kibuule, Amos Massele, Ibrahim Chikowe, Felix Khuluza, Tinotenda Taruvinga, Abubakr Alfadl, Elfatih Malik, Margaret Oluka, Sylvia Opanga, Daniel N. A. Ankrah, Israel A. Sefah, Daniel Afriyie, Eunice T. Tagoe, Adefolarin A. Amu, Mlungisi P. Msibi, Ayukafangha Etando, Mobolaji E. Alabi, Patrick Okwen, Loveline Lum Niba, Julius C. Mwita, Godfrey M. Rwegerera, Joyce Kgatlwane, Ammar A. Jairoun, Chioma Ejekam, Rooyen T. Mavenyengwa, Irene Murimi-Worstell, Stephen M. Campbell, Johanna C. Meyer

**Affiliations:** 1Department of Pharmacology, Therapeutics and Toxicology, Lagos State University College of Medicine, Lagos 100271, Nigeria; 2Department of Medicine, Lagos State University Teaching Hospital, Lagos 100271, Nigeria; 3Department of Pharmacoepidemiology, Strathclyde Institute of Pharmacy and Biomedical Sciences, University of Strathclyde, Glasgow G4 0RE, UK; 4Centre of Medical and Bio-Allied Health Sciences Research, Ajman University, Ajman 346, United Arab Emirates; 5Department of Public Health Pharmacy and Management, School of Pharmacy, Sefako Makgatho Health Sciences University, Pretoria 02084, South Africa; 6Department of Pharmacology and Therapeutics, Ekiti State University, Ado Ekiti 362103, Nigeria; 7Department of Medicine, Ekiti State University Teaching Hospital, Ado Ekiti 360211, Nigeria; 8Department of Pharmacy, School of Health Sciences, University of Zambia, Lusaka P.O. Box 50110, Zambia; 9Department of Disease Control, School of Veterinary Medicine, University of Zambia, Lusaka P.O. Box 32379, Zambia; 10Nigerian Center for Disease Control, Plot 801, Ebitu Ukiwe Street, Jabi, Abuja 900108, Nigeria; 11Department of Pharmacy, Lagos State University Teaching Hospital, Lagos 100271, Nigeria; 12Child, Youth and School Health Directorate, National Department of Health, Pretoria 0083, South Africa; 13Children’s Cancer Hospital, Egypt-57357 (CCHE-57357), Cairo 11441, Egypt; 14Biomedical Research Department, Armed Forces College of Medicine, Cairo 11774, Egypt; 15CIMAS, Cimas House, Borrowdale Office Park, Borrowdale Road, Harare P.O. Box 1243, Zimbabwe; 16Department of Medicine, University of Zimbabwe College of Health Sciences, Harare P.O. Box MP167, Zimbabwe; 17Department of Child Health and Paediatrics, Egerton University, Nakuru P.O.Box 536, Kenya; 18East Africa Centre for Vaccines and Immunization (ECAVI), Namela House, Naguru, Kampala P.O. Box 3040, Uganda; 19Department of Pharmacology & Therapeutics, Busitema University, Mbale P.O. Box 236, Uganda; 20Department of Clinical Pharmacology and Therapeutics, Hurbert Kairuki Memorial University, 70 Chwaku Road Mikocheni, Dar Es Salaam P.O. Box 65300, Tanzania; 21Pharmacy Department, Formerly College of Medicine, Kamuzu University of Health Sciences (KUHeS), Blantyre P.O. Box 278, Malawi; 22Department of Global Health and Development (GHD), London School of Hygiene and Tropical Medicine (LSHTM), London WC1E 7TH, UK; 23National Medicines and Poisons Board, Federal Ministry of Health, Khartoum P.O. Box 303, Sudan; 24Department of Pharmacy Practice, Unaizah College of Pharmacy, Qassim University, Unaizah 51911, Saudi Arabia; 25Department of Community Medicine, Faculty of Medicine, University of Khartoum, Khartoum 11111, Sudan; 26Department of Pharmacology & Pharmacognosy, School of Pharmacy, University of Nairobi, Nairobi P.O. Box 19676-00202, Kenya; 27Department of Pharmaceutics and Pharmacy Practice, School of Pharmacy, University of Nairobi, Nairobi P.O. Box 19676-00202, Kenya; 28Department of Pharmacy, Korle Bu Teaching Hospital, Accra P.O. Box 77, Ghana; 29Pharmacy Practice Department, School of Pharmacy, University of Health and Allied Sciences, Hohoe PMB 31, Ghana; 30Pharmacy Department, Ghana Police Hospital, Accra P.O. Box CT104, Ghana; 31Department of Management Science, University of Strathclyde, Glasgow G4 0QU, UK; 32Pharmacy Department, Eswatini Medical Christian University, P.O. Box A624, Swazi Plaza, Mbabane H100, Eswatini; 33Faculty of Health Sciences, Department of Medical Laboratory Sciences, Eswatini Medical Christian University, Swazi Plaza P.O. Box A624, Mbabane H100, Eswatini; 34Department of Medical Laboratory Sciences, Faculty of Health Sciences, Eswatini Medical Christian University, Swazi Plaza P.O. Box A624, Mbabane H100, Eswatini; 35School of Pharmaceutical Sciences, College of Health Sciences, University of Kwazulu-natal (UKZN), Durban 4001, South Africa; 36Effective Basic Services (eBASE) Africa, Ndamukong Street, Bamenda 5175, Cameroon; 37Faculty of Health and Medical Sciences, Adelaide University, Adelaide 5005, Australia; 38Department of Public Health, University of Bamenda, Bambili P.O. Box 39, Cameroon; 39Department of Internal Medicine, Faculty of Medicine, University of Botswana, Gaborone P.O. Box 70480, Botswana; 40Department of Medicine, Sir Ketumile Masire Teaching Hospital, Gaborone P.O. Box 70480, Botswana; 41Department of Pharmacy, University of Botswana, Gaborone P.O. Box 70480, Botswana; 42Health and Safety Department, Dubai Municipality, Dubai P.O. Box 67, United Arab Emirates; 43School of Pharmaceutical Sciences, Universiti Sains Malaysia, Penang 11800, Malaysia; 44Department of Community Health, Lagos University Teaching Hospital, Idi-Araba, Lagos PMB 21266, Nigeria; 45Medical Microbiology Unit, Faculty of Medicine and Health Sciences, University of Zimbabwe, Harare P.O. Box MP167, Zimbabwe; 46School of Pharmacy, Massachusetts College of Pharmacy and Health Sciences, Boston, MA 02115, USA; 47Centre for Epidemiology and Public Health, School of Health Sciences, University of Manchester, Manchester M13 9PL, UK; 48NIHR Greater Manchester Patient Safety Translational Research Centre, School of Health Sciences, University of Manchester, Manchester M13 9PL, UK

**Keywords:** COVID-19, vaccination, hesitancy, availability, challenges, African countries, policy implications, social media

## Abstract

The introduction of effective vaccines in December 2020 marked a significant step forward in the global response to COVID-19. Given concerns with access, acceptability, and hesitancy across Africa, there is a need to describe the current status of vaccine uptake in the continent. An exploratory study was undertaken to investigate these aspects, current challenges, and lessons learnt across Africa to provide future direction. Senior personnel across 14 African countries completed a self-administered questionnaire, with a descriptive analysis of the data. Vaccine roll-out commenced in March 2021 in most countries. COVID-19 vaccination coverage varied from low in Cameroon and Tanzania and up to 39.85% full coverage in Botswana at the end of 2021; that is, all doses advocated by initial protocols versus the total population, with rates increasing to 58.4% in Botswana by the end of June 2022. The greatest increase in people being fully vaccinated was observed in Uganda (20.4% increase), Botswana (18.5% increase), and Zambia (17.9% increase). Most vaccines were obtained through WHO-COVAX agreements. Initially, vaccination was prioritised for healthcare workers (HCWs), the elderly, adults with co-morbidities, and other at-risk groups, with countries now commencing vaccination among children and administering booster doses. Challenges included irregular supply and considerable hesitancy arising from misinformation fuelled by social media activities. Overall, there was fair to reasonable access to vaccination across countries, enhanced by government initiatives. Vaccine hesitancy must be addressed with context-specific interventions, including proactive programmes among HCWs, medical journalists, and the public.

## 1. Introduction

In March 2020, SARS-CoV-2, the virus that caused the coronavirus disease of 2019 (COVID-19), was declared a pandemic by the World Health Organisation (WHO) [1], and by late June 2022, there were 540 million confirmed cases of COVID-19 globally, with over 6.3 million deaths recorded [2].

The focus among countries and continents, certainly initially, was the introduction of public health policies to try and slow down the spread of the virus, with its subsequent impact on morbidity and mortality, in the absence of proven treatments and vaccines to treat COVID-19 [3,4]. These policies included lockdown measures incorporating the closure of educational establishments and borders, promoting hand hygiene, social distancing and the wearing of personal protective equipment (PPE) as well as quarantining measures [5,6,7,8,9,10,11,12,13]. However, there was variable implementation and adherence to the recommended preventative measures across countries, which adversely affected the subsequent prevalence and mortality rates [14,15,16,17,18]. 

Several re-purposed medicines were proposed for the prevention and management of patients with COVID-19 in the absence of effective vaccines. These included hydroxychloroquine, lopinavir/ritonavir, ivermectin, remdesivir, and steroids [19,20,21,22,23,24], with their endorsement resulting in appreciably increased utilisation, especially hydroxychloroquine, fuelled by social media and other activities [16,20,25,26,27,28]. This surge was despite limited evidence regarding their effectiveness, apart from dexamethasone, initially and in subsequent studies, with their overuse increasing morbidity, mortality, and costs [16,26,29,30,31,32,33,34,35,36,37]. These concerns resulted in calls across countries to enhance the evidence base of treatments before they were routinely recommended, thereby minimising the potential for misinformation [28,38,39,40,41]. 

Alongside this, the unintended consequences of lockdown and social distancing measures, including limited access to healthcare services, were considerable, especially in low- and middle-income countries (LMICs), including African countries [13,42,43,44,45,46,47,48,49,50,51,52,53]. The unintended consequences also included increased morbidity and mortality from reduced routine vaccinations among children in Africa [54,55,56,57]. 

Consequently, there was an appreciable need for effective vaccines to limit the spread of the virus. Numerous published studies have demonstrated the effectiveness of COVID-19 vaccines in reducing the impact of COVID-19 across countries, including reducing mortality, especially for patients at risk of severe disease [58,59,60,61]. These effectiveness rates resulted in a generally high acceptance of COVID-19 vaccines when available across countries [62], with booster campaigns introduced to tackle new variants and the waning of vaccine effectiveness over time [60,63,64]. However, there have been concerns with the vaccines across countries increasing hesitancy [65].

High acceptance rates (up to 88.8% acceptance with a 95% effectiveness rate) for COVID-19 vaccines were seen in a study by Bono et al., (2021) involving LMICs, including five African countries [66], although they were lower (61%) in the pooled study of Norhayati et al., (2022) [67]. Kanyanda et al., (2021) also generally identified high acceptance rates for the vaccine across sub-Saharan Africa, although they were lower in Mali (64.5%) [68]. Norhayati et al., (2022) also showed an acceptance rate of only 53% among the 15 African countries in their systematic review [67]. However, high acceptance rates were seen among the public in Nigeria, ranging from 74.5% to 85.3% of those surveyed [69,70,71], although they were lower in the study by Tobin et al., (2021) at 50.2% [72]. The major reasons for the non-acceptance of COVID-19 vaccines in Nigeria included concerns with the robustness of the published clinical trials, including the length of the follow-up when first rolled-out and the age of the included patients in the trials [69,71].

However, as with the increasing administration of COVID-19 vaccines, concerns regarding some of the rare adverse effects of the vaccines have contributed to vaccine hesitancy [73,74,75,76,77], with vaccine hesitancy defined as ‘a delay in acceptance or refusal of vaccination, despite the availability of vaccination services’ [78,79]. These concerns have resulted in increased hesitancy towards the COVID-19 vaccines across countries, including African countries [65,80,81,82,83]. Across Africa, studies have documented that between 32–37% of adults would not accept the vaccine, with hesitancy rates influenced by age, education, source of information, income and/or employment status, and the potential for increased infection [84,85,86,87]. Variable acceptance rates were also seen among African countries in the study of Sallam et al., (2022) [80], with variable hesitancy between 21% to 84.6% of those surveyed also seen in Cameroon, Ghana, Kenya, South Africa, Zimbabwe, and Zambia [82,86,88,89,90,91,92]. Whilst there have been challenges with vaccine hesitancy in Zimbabwe when COVID-19 vaccines were first made available, this was reduced with national and local community engagement programmes [93].

In Tanzania, the Health Minister in early 2021 stated that the country would not partake in vaccination campaigns as they were not satisfied with the safety of the vaccines, relying on traditional and household herbs and medicines for prevention and treatment [94]. Whilst this situation changed later in the year, appreciable hesitancy remained [95]. Alongside this, there have also been concerns with hesitancy among healthcare workers (HCWs), including healthcare professionals (HCPs), and students across Africa [96,97,98].

COVID-19 vaccine hesitancy is a key issue to address, with vaccine hesitancy already in 2019 identified by the WHO as one of the top ten global threats to public health [99,100]. Overall, a considerable number of deaths could have been averted if target vaccination rates had been achieved, especially among low- and middle-income countries, including African countries [101]. As mentioned, key attributes among those hesitant to COVID-19 vaccines include age, level of education, income and/ or employment status, and locality [84,86,87,102,103,104,105]. Religious beliefs and political issues are also key areas influencing hesitancy across Africa [106]. Identifying key reasons regarding vaccine hesitancy is important among African countries given the documented effectiveness of the vaccines, their high rates of infectious diseases, as well as high rates of antimicrobial resistance (AMR) exacerbated by excessive use of antibiotics to treat patients with COVID-19 [93,107,108,109,110,111,112,113].

Identified concerns to address include confidence surrounding the vaccines, including their effectiveness and potential safety issues, as well as addressing complacency issues incorporating beliefs of a low risk of catching COVID-19 and a low disease severity if COVID-19 is caught [65,114,115,116,117,118]. Enhancing access (convenience) and instigating robust communication programmes adjusted to the socio-demographics of the target population (context) are also important to address misinformation and disinformation promulgated via social media [103,114,119,120]. Addressing COVID-19 vaccine hesitancy is also important for the acceptance of other vaccines, as well as helping to address high AMR rates across Africa [111,121,122].

Other important challenges affecting the availability and use of COVID-19 vaccines include the availability of supplies and trained HCWs, including HCPs, to administer the vaccines once available [123].

Consequently, there is a need to build on these studies. This includes documenting key issues regarding COVID-19 vaccines across Africa, including their acceptance and challenges. Subsequently, documenting key activities that can be undertaken by governments and HCPs to address hesitancy to improve future vaccination rates for this and future pandemics.

## 2. Materials and Methods

### 2.1. Study Design

A mixed methods approach was adopted. This is similar to other Pan-African projects undertaken by the co-authors to document and debate key issues surrounding both non-infectious diseases and infectious diseases, as well as general areas, to provide future guidance [10,15,26,124,125,126,127,128,129,130]. The first stage comprised a narrative review of the literature regarding the effectiveness and safety of current vaccines for COVID-19, along with acceptance rates and hesitancy across Africa and the reasons for this. As mentioned, hesitancy was defined as ‘a delay in acceptance or refusal of vaccination, despite the availability of vaccination services’ [78,79,131]. The principal objective was to derive key discussion points for the second stage of the research. This was not a systematic review since the principal aim of this paper was to document the current situation regarding the vaccines, including vaccine hesitancy and the challenges among sub-Saharan African countries to provide future direction. The literature review was largely based on the considerable knowledge of the senior-level co-authors. This included individual country studies documenting current vaccination and hesitancy rates known to the co-authors from each country, as well as Pan-African and Global studies discussing similar issues. We adopted this approach before when discussing key activities and their future implications across countries and continents including Africa, with the deliberations based on the considerable knowledge and experience of the senior-level co-authors [125,126,127,128,129,130,132,133].

The second part of the study comprised an explorative questionnaire survey among senior-level government, HCP, and academic personnel from a range of African countries. The countries were purposefully selected based on the availability and knowledge of the senior-level co-authors to address the key issues and objectives of the paper. An analytical framework approach was used alongside a pragmatic paradigm aimed at providing future guidance, including for future pandemics [134,135,136,137]. The participating countries (Table 1) provided a range of economic status (Gross Domestic Product (GDP)/capita) [138], population size [139], and geographies, as well as current infection and mortality rates [2], to meet study objectives. We are aware though that there can be concerns with reporting mortality rates, including definitions [140,141,142].

### 2.2. Questionnaire Design and Analysis

The key questions posed to participating countries following a narrative review of the literature included the following:Did your country have a dedicated COVID-19 vaccine rollout programme? Was this in the public sector, private sector, or both, and were any specific age groups covered?Which COVID-19 vaccines were made available and how were the costs covered for each (e.g., NGOs)?What is the current coverage rate (different doses if known)?What is being done to ensure access to COVID-19 vaccines, and how would you describe the acceptance (willingness) of the population to COVID-19 vaccinations?What is the extent of any misinformation about the vaccine (if known), and how is misinformation being spread (e.g., social media)?What are the challenges with COVID-19 vaccinations in addition to the above, and what are potential ways forward or measures being undertaken by national authorities and other key stakeholders to mitigate against these challenges?

Within each country, the co-authors collated the replies, which were subsequently reviewed and collated by the principal author (OO). The findings were then fed back to each country for clarification to enhance their accuracy. A common basis was used to compare vaccine findings across countries, building on country-specific information [143,144,145]. The final responses were subsequently analysed using thematic analysis techniques [10,146].

Common themes from the responses were identified and discussed with the co-authors to provide future guidance [10]. The findings were subsequently summarised into key themes and challenges faced by participating countries. Potential ways forward were broken down into the four Es where pertinent, namely ‘Education, Engineering, Economics and Enforcement’ [147,148], in order to consolidate approaches. We have used this methodology before when consolidating potential approaches and activities across disease areas to improve the use of medicine [124,127,149,150]. ‘Education’ includes disseminating information to key stakeholder groups and developing guidelines or formularies [151,152,153,154]. ‘Engineering’ includes organisational or managerial interventions such as instigating and monitoring prescribing targets and quality targets [148,150]. ‘Economics’ include financial incentives to key stakeholder groups and ‘Enforcement’ includes regulations by law including the banning of self-purchasing of antibiotics without a prescription [148,155,156].

Two timescales were employed to assess changes over time as more knowledge became available regarding the effectiveness and safety of the vaccines including boosters. These were up to the end of December 2021 and up to the end of June 2022.

### 2.3. Ethical Considerations

No ethical approval was sought for this study as no human subjects were involved. In addition, the co-authors, who were very knowledgeable in their country concerning these matters, voluntarily provided the information, which are typically available in the public domain. This mirrors similar studies conducted by the co-authors across Africa and wider, involving general subjects as well as both infectious and non-infectious diseases, and is in accordance with institutional guidance [10,15,26,124,125,126,127,129,157,158].

## 3. Results

We will first document the initial sources of COVID-19 vaccines across Africa before discussing initial and subsequent coverage rates as well as key issues surrounding access, hesitancy and challenges.

### 3.1. Vaccine Sources and Deployment

Vaccine roll-out commenced in a number of African countries in the first quarter of 2021, with Egypt the first African country among those studied to commence vaccination on 24 January 2021, followed by South Africa on 17 February 2021 and Zimbabwe on 18 February 2021. Other countries studied, apart from Tanzania, introduced their COVID-19 vaccines between March and April 2021 [94]. Most of the countries had dedicated vaccine roll-out programmes involving both the public and private sectors (Appendix A).

The sources of vaccines among the countries were typically from donations by multilateral agencies, non-governmental organisations, and higher income economies, including the UK and USA, with some countries, including South Africa, entering into bilateral agreements (Appendix A). Agencies and other organisations included the COVAX-WHO initiative, the African Union Vaccine Acquisition Trust (AVAT) and GAVI, the Vaccine Alliance, and the Serum Institute of India. COVAX is co-led by the Coalition for Epidemic Preparedness Innovations (CEPI), GAVI, the Vaccine Alliance, and the WHO, alongside a key delivery partner, UNICEF [159]. Typically, vaccines from multiple sources were administered across Africa (Appendix A).

### 3.2. Vaccination Coverage

As of 31 December 2021, vaccination coverage in the studied countries varied from very low rates in Tanzania and Nigeria, with higher rates reported in Botswana, Egypt, Eswatini, and South Africa (Table 2 and Table 3). Most of the studied African countries deployed vaccination programmes in phases, prioritising HCWs, followed by elderly patients and patients with co-morbidities at high risk of severe COVID-19 disease, hospitalisation, or death should they be infected with SARS-CoV-2 (Appendix A). A number of these countries also commenced the vaccination of children of certain age groups and began administering booster doses to the adult population by the end of December 2021 (Appendix A).

The most widely administered vaccines by the end of June 2022 across Afria were Johnson & Johnson (30.3% of the total number administered), Pfizer BioNtech (19.1%), Sinopharm (14.2%), Oxford AstraZeneca (13.7%), Sinovac (7.9%), and Moderna (5.5%) [143].

By the end of June 2022, COVID-19 vaccination coverage had increased across Africa, with an overall 18.4% full vaccination coverage, up from 8.9% at the end of December 2021. This was considerably lower though than the global rate of 60.7%, up from 49.2% in December 2021. However, again, appreciable variation was observed across the various countries (Table 2 and Table 3), with the greatest increases seen in Uganda (20.4%), Botswana (18.5%), Zambia (17.9%), and Ghana (14.8%) and with varied increase in coverage seen in the other studied countries (Table 2). Figure 1 depicts the overall increase in the administration of COVID-19 vaccines by the end of June 2022. The lowest coverage rate among the studied countries was recorded in Cameroon, with only 4.5% of the population being fully vaccinated by mid-2022.

### 3.3. Access, Hesitancy, and Challenges with the COVID-19 Vaccine Roll-Out

Table 4 summarises the levels of access, acceptance, hesitancy, and challenges with the COVID-19 vaccine roll-out across Africa, which builds on Pan-African and other African studies [66,67,68,69,70,71,72,80,81,82,83,84,85,86]. Access was generally well facilitated by the different measures and initiatives among the various African governments; however, there were concerns in some African countries, including the variable availability of the different vaccines.

There were also concerns with the level of acceptance for the vaccine among a number of the studied countries, with subsequent high rates of vaccine hesitancy in some of these. In Cameroon, poor acceptance levels were observed, and Egypt and Eswatini initially presented with low levels of acceptance; however, there were considerable improvements over time.

Misinformation concerning the side-effects of the vaccine coupled with other key issues, including fertility and other conspiracy theories, were widely circulated on social media platforms. This resulted in issues of trust and high hesitancy rates among some of the studied countries. Addressing these and other highlighted challenges will be key to improving vaccination rates in Africa going forward.

### 3.4. Lessons Learnt and Ways Forward

A considerable number of lessons learned and ways forward to improve future vaccination rates were identified across countries. These are summarised in Table 5 and include increasing trust in governments and other key stakeholder groups, including HCPs, which has been eroded with increasing vaccine hesitancy rates [199]. This involves reducing doubts about the vaccines, including COVID-19 vaccines, among HCWs including HCPs [175,199,200,201], through social media and other channels, with social media playing an increasing role in promulgating misinformation [120,202]. Trusted politicians endorsing COVID-19 vaccines can also reduce hesitancy [203].

## 4. Discussion

We believe this is the first study to comprehensively review current African vaccination and hesitancy rates alongside current challenges, as well as propose potential ways forward given current concerns. This is particularly important in Africa with high existing rates of infectious diseases as well as high rates of AMR [111,121,130]. The overuse of antibiotics in treating patients with COVID-19 has further enhanced AMR rates, which urgently need to be addressed to reduce future morbidity, mortality, and costs [112,113,124,227]. The majority of African countries started their roll-out of COVID-19 vaccines in early 2021, in many cases with vaccines donated as part of COVAX, GAVI, AVAT, or bilateral agreements for specific countries (Appendix A). Tanzania was the last of the African countries to initiate its vaccine roll-out due to denial initially [94]. However, some of the COVID-19 vaccines donated were near their expiration date, creating additional challenges.

There was appreciable growth in vaccination rates in a number of the African countries surveyed between the end of 2021 and mid-2022, enhanced by greater availability and access, with Uganda, Botswana, and Zambia recording the greatest increase in vaccination coverage rates (Table 2, Figure 1). However, coverage rates continued to remain relatively low in some of the African countries surveyed, including Cameroon, Nigeria, Sudan, and Tanzania (Table 1), exacerbated by high hesitancy rates fuelled by misinformation (Table 4). These low rates impacted the global coverage rate of 60.7% in mid-year of 2022, which is below the target of 70% set by the WHO [228]. Botswana and Egypt continued to have high coverage rates among the African countries surveyed, enhanced by pro-active activities by regional and national governments (Table 2 and Table 4). The multi-faceted activities they undertook provide guidance to other African countries, including measures used to ensure availability among the general population.

There are still concerns with low vaccination rates of children in a number of the surveyed countries, with only 7% of doses administered among 23 African countries by the end of June 2022 to children and adolescents younger than 18 years of age [229]. This low vaccination rate may have been exacerbated by supply challenges as well as concerns about the safety of the vaccines among children. We will be following this up in future studies, as concerns with vaccinating children with the COVID-19 vaccines may negatively impact routine immunisation programmes among children, which were already severely compromised by the pandemic [54,55,230].

Hesitancy continued to be a concern across Africa. This is fuelled by the level of misinformation including false claims about its side-effects, the disease mainly targeting rich urban populations, it being the disease of the west, it likely to be only a mild disease among Africans, it interfering with fertility, and the ‘mark of the beast’, exacerbated by social media activities [120,215] (Table 4).

Several activities were identified to improve vaccination rates during the current and future pandemics (Table 5). These include the greater involvement of African countries in basic research and clinical research, epidemiology, as well as improving current pharmacovigilance activities, which are a concern across Africa [205,206,231]. In addition, these include improving the education of HCPs during undergraduate training and post-qualification using a variety of hybrid approaches, with hybrid learning here to stay post-pandemic [10]. Community pharmacists can also play a key role in improving vaccination rates. This is because they are often the first point of contact between patients and HCPs, particularly in rural areas and where there are high patient co-payments [208,209,210]. Alongside this, governments should make the vaccines easily available to reduce travelling times and associated costs, which could negatively impact uptake. The government and key HCPs also need to actively engage in social media activities [120]. This is because misinformation can rapidly circulate via social media, with significant implications for trust in governments as well as the prevention and management of COVID-19 if this continues [20,120,175,215]. Compulsory vaccination of certain groups has been instigated in some countries to reduce transmission rates. However, before doing this, governments need to carefully consider legal, ethical, and other major issues [225,226].

We will continue to monitor key areas across Africa regarding current vaccination rates and challenges, including continued hesitancy. This will enable African countries to continue to learn from each other. As a result, the impact of the virus can be reduced, which includes the unintended consequences of measures that were initially implemented to contain the pandemic. In addition, the continued high inappropriate use of antibiotics across Africa needs to be reduced, building on current national action plans to reduce AMR [124,130].

## 5. Conclusions

It is widely acknowledged that the introduction of COVID-19 vaccines was a significant step forward across countries to reduce the morbidity, mortality, and costs associated with the virus. This includes the unintended consequences of public health measures instigated to reduce its spread and impact. There were appreciable variations in vaccine coverage and hesitancy across Africa. This was fuelled by critical issues, including access, irregular supply, and the level of misinformation circulating within communities and on various media platforms. Key among these included was misinformation fuelled by social media. A number of activities were identified to address this situation across Africa, which included instigating improved pharmacovigilance activities and addressing the negative impact of social media, which will be followed up in future studies.

## Figures and Tables

**Figure 1 vaccines-10-01553-f001:**
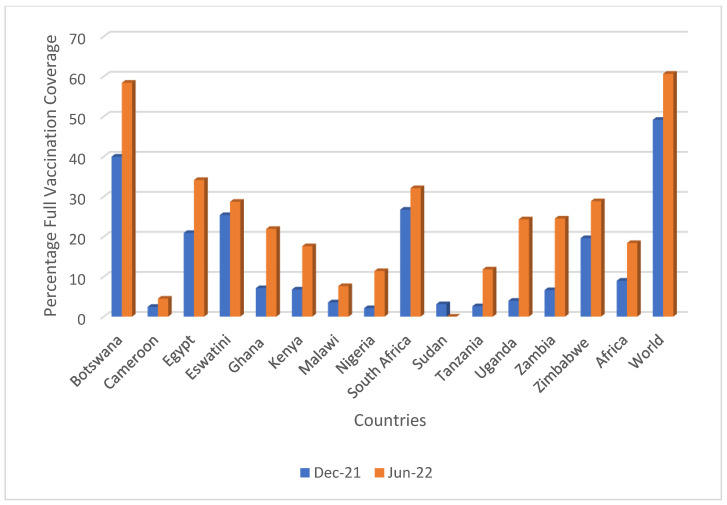
Changes in full vaccination coverage between December 2021 and June 2022. NB: DEC-21-F = fully vaccinated by protocol as of 31 December 2021, JUN-22-F = fully vaccinated by the protocol as of 30 June 2022. Based on [143,144,145,160].

**Table 1 vaccines-10-01553-t001:** Current population size, GDP/capita, and COVID-19 infection rates among participating African countries.

Country	Population Size (Thousands)	GDP/Capita (US$)	Accumulated Infection Rate (Thousands)	Accumulated Mortality Rate (Thousands)
Botswana	2351.63	6711.0	325.5	2.77
Cameroon	26,545.86	1499.4	121.0	1.93
Egypt	102,260.0	4028.4	515.3	1.93
Eswatini	1160.16	3415.5	73.3	1.42
Ghana	31,072.94	2328.5	168.5	1.46
Kenya	53,771.30	1838.2	338.1	5.67
Malawi	19,129.95	625.3	87.7	2.67
Nigeria	206,139.60	2097.1	263.1	3.15
South Africa	59,308.69	5090.7	4010.2	102.1
Sudan	48,892.81	595	63.2	4.96
Tanzania	61,498.44	1136	38.7	0.84
Uganda	47,123.53	858	168.7	3.63
Zambia	18,383.96	1050.9	332.5	4.02
Zimbabwe	14,862.92	1128.2	256.6	5.59

**Table 2 vaccines-10-01553-t002:** COVID-19 vaccination coverage across African countries as of 31 December 2021 and 30 June 2022.

Country	Vaccination Coverage—31 December 2021—% of the Total Population		Vaccination Coverage—30 June 2022—% of the Total Population	
	Full (Completed)	Partial	Full (Completed)	Partial
Botswana	39.9	5.2	58.4	7.1
Cameroon	2.4	0.6	4.5	1.3
Egypt	20.9	11.5	34.1	12.0
Eswatini	25.4	2.7	28.7	5.7
Ghana	7.1	10.5	21.9	9.9
Kenya	6.8	4.4	17.6	6.3
Malawi	3.5	3.96	7.63	3.05
Nigeria	2.1	2.7	11.4	5.6
South Africa	26.7	5.2	32.1	5.0
Sudan	3.1	3.7	10.4	NA
Tanzania	2.3	0.7	11.8	2.2
Uganda	3.9	18.8	24.3	11.1
Zambia	6.6	NA	24.5	35.0
Zimbabwe	19.6	6.2	28.8	10.6
Africa	8.9	4.9	18.4	5.2
World	49.2	8.5	60.7	5.5

NB: Data sources—[143,144,145,160]. NB: Full coverage means the total number of people who received all doses prescribed by the initial vaccination programme divided by the total population of the country [144].

**Table 3 vaccines-10-01553-t003:** Number of vaccine doses utilised by the countries as of 31 December 2021 and 30 June 2022.

Countries	Doses as of 31 December 2021	Doses as of 30 June 2022
Botswana	NA	2.73 million
Cameroon	1.02 million	1.85 million
Egypt	57.49 million	91.45 million
Eswatini	404,374	684,176
Ghana	7.76 million	18.24 million
Kenya	10.12 million	18.54 million
Malawi	1.8 million	3.17 million
Nigeria	14.84 million	55.47 million
South Africa	27.97 million	36.82 million
Sudan	3.64 million	NA
Tanzania	2.43 million	12.07 million
Uganda	12.09 million	21.76 million
Zambia	1.73 million	7.2 million
Zimbabwe	7.26 million	11.97 million
Africa	303.51 million	550.21 million
World	9.18 billion	12.1 billion

NB: Data sources—[144,145,160].

**Table 4 vaccines-10-01553-t004:** Levels of access, acceptance, hesitancy, and challenges with COVID-19 vaccines across Africa.

Country	Access, Acceptance, Hesitancy and Challenges
Botswana [161,162]	**Access**: Typically, good and easy access, with District Health Management Teams (DHMT) opening vaccination sites across Botswana. Access was also improved by involving selected private facilities. Drive-through COVID-19 vaccination campaigns at the University of Botswana, Gaborone, and in Francistown also improved access. **Acceptance**: High acceptance rates as evidenced by people travelling some distance to seek vaccinations potentially enhanced by most people in Botswana either having been previously affected by COVID-19 or having lost a close family member/friend to COVID-19. A deployment plan was undertaken a few weeks before the first vaccine arrived, and the survey showed an acceptance rate of 76%. **Hesitancy**: Some hesitancy was identified in an early survey at the beginning of the vaccination rollout program among younger HCWs exacerbated by the spread of misinformation; however, this is changing. Initially, social media was the main source of information generally across Botswana, which reported mainly selected AEFIs. The government’s Task Force counteracted that by providing updates on COVID-19 and vaccinations, as well as campaigns such as ‘Arm Ready’. The Botswana Medicines Regulatory Authority was also involved in sharing information on vaccinations and AEFIs. **Challenges**: Shortage of vaccinators, HCPs, and support staff. Equipment shortages and low internet bandwidth in some areas hampering the use of the electronic data systems to track the administration of vaccines and adverse effects.
Cameroon [82,143,144]	**Access**: Access was high with availability across all health facilities. **Acceptance**: Typically, poor acceptance reflected in high levels of hesitancy in Cameroon. **Hesitancy**: High hesitancy because of mistrust of the vaccine and the ongoing myth that the COVID-19 is not a serious disease in black Africans. Considerable spread of misinformation and propaganda such as “you can light a bulb from vaccination sites on the skin”, “plot to use genetic coded molecule to make humans barren through vaccination”. **Challenges**: Difficulties with accessing hard to reach locations for vaccination. Poor storage conditions with frequent power outages. Limited human resources to cover a large population. Successfully addressing the misinformation and increasing trust in the COVID-19 vaccines.
Egypt [85,163,164,165,166]	**Access**: Access appears easy for the whole population, with the Egyptian Ministry of Health and Population deploying a web-based vaccine management system active in effect on March 2021. This system allows citizens to register with their national passports and mobile numbers and to receive notifications on assigned date, time, and place for vaccination. In mid-September 2021, the Egyptian Prime Minister launched a campaign called “Together To Reassure” to encourage citizens to register on the Health Ministry’s website to obtain their COVID-19 vaccines in three governorates (Cairo, Giza, and Alexandria) in the initial stage and later in other governorates. The campaign lasted 10 days from 10 h to 22 h and included three open-topped and equipped buses touring each governorate. **Acceptance**: The acceptance of the COVID-19 vaccine has been variable across Egypt. Based on a cross-sectional study undertaken among medical students in Egypt, 90.5% of study participants recognised the importance of COVID-19 vaccination, but 46% were vaccine hesitant. **Hesitancy**: Vaccine hesitancy has been reported in many studies across Egypt due to a number of reasons. These include: Signing informed consent before vaccination gives the perception to people that they will be part of a clinical trial, thereby increasing public fear and refusal. A false belief that the COVID-19 vaccination would be offered to only a specific segment of the population. False claims on social media about AEFIs and including information/misinformation about patients who died after their vaccination. Circulation of conspiracy theories on various social media platforms. **Challenges**: Reports of blood clotting in people after their vaccination increased public panic and hesitancy. There are concerns especially among highly educated people regarding the safety and efficacy of the vaccines. Other challenges include limited data regarding the COVID-19 vaccine’s adverse effects, the perceived unsafe nature of the vaccine, and fear of genetic alterations (especially the Pfizer vaccine). Effectively addressing misinformation regarding COVID-19 vaccines circulating in various social medica platforms and increasing trust in the vaccines where there are concerns.
Eswatini [167,168]	**Access**: Access is good, decentralization to the ‘Tinkhundla’ centres (Chiefdoms) bringing services to the rural population eliminating traveling cost. Vaccination is also available in shopping malls and all public centres. **Acceptance**: Acceptance was low at the beginning of the vaccination program but has gradually improved with increasing educational and awareness campaigns. The increase in fatalities from COVID-19 has also made the public increasingly consider the vaccine. Awareness was strengthened and acceptance was improved through public media campaigns with the sharing of evidence-based information on vaccine safety and differential mortality between vaccinated and unvaccinated groups. Public figures, e.g., governors, religious leaders, and celebrities, aired receiving vaccines and testifying to their safety and benefits. **Hesitancy**: Vaccine hesitancy was exacerbated in Eswatini by misinformation and myths, especially on social media alongside social and religious beliefs. This included fake news relating to the level of side-effects, speculated alteration of the DNA sequence, and an alleged plan of the G5 network to reduce the world population. There are ongoing activities to try and address this. These include strengthening awareness through social and public media regarding the effectiveness and safety of current vaccines alongside the morbidity and mortality associated with COVID-19 among unvaccinated patients. In addition, potentially posting documentaries as well as testaments from key personnel including HCPs, religious leaders, and others regarding the benefits of the vaccine to the population. **Challenges**: Speculations that the vaccine design might not be effective against all the possible variants or mutations. Lack of mutational analysis or genetic sequences to map the genetic variants and mutations currently in circulation. Non-compliance with pharmacological interventions alongside continued vaccine hesitancy could lead to further lockdowns. This needs to be avoided going forward through proactive communications via social media and other platforms.
Ghana [15,92,169,170,171,172,173,174]	**Access**: COVID-19 vaccination is currently free in Ghana, but access to vaccination has been a challenge due to frequent shortages. **Acceptance**: Different acceptability levels among different populations in Ghana, including HCPs. Vaccine acceptance in Ghana ranges from 41% to 71% of the surveyed population, according to several studies undertaken in Ghana. **Hesitancy**: Hesitancy has been observed in Ghana as a threat to controlling the pandemic among the general population, which needs to be addressed. The most common reasons for vaccine hesitancy include misinformation through the media, fear of harmful side-effects, and mistrust of the authorities. **Challenges**: Vaccine availability is a real challenge for Ghana, as quantities are not sufficient at vaccination points to ensure continuous vaccination. In addition, there is a need for pro-active approaches form key stakeholder groups, including Ministry of Health personnel, to address the misinformation that is exacerbating vaccine hesitancy as well as restore trust in the government. Ghana was very proactive with a number of public health and other measures at the start of the pandemic, which needs to continue.
Kenya [120,175,176,177,178,179]	**Access**: The Ministry of Health (MoH) website is full of information urging citizens to get vaccinated, with increased availability of different types of vaccines so that citizens can choose. Increased use of public places including markets, churches, mosques, bus stops, and other public areas for vaccination so that a variety of HCWs can reach out to the population—with variable acceptance among community health volunteers. Vaccination production/manufacture has been introduced in Kenya to supplement donated vaccines. **Acceptance**: There are concerns with current acceptance rates. In August 2021, 36% of Kenyans surveyed were unsure about COVID-19 vaccines due to conflicting information, rumours, misinformation, and conspiracy theories. Social and mainstream media was used to try and address this, building on the increasing role of social media worldwide with influencing vaccine hesitancy coupled with the need to address issues of trust, with information emanating from the MoH and other key stakeholder groups. The government also suggested restricting the access of certain services to citizens who are not fully vaccinated. **Hesitancy**: Vaccine hesitancy exists and was estimated at 15% in March 2021 before the vaccine roll out program. The principal reasons for hesitancy include beliefs that vaccines will affect the reproductive system; people in rural areas do not believe COVID 19 is real; anti-vaxxers on social media (typically outside of Kenya); and concerns with the safety of current vaccines. Conspiracy theories, e.g., vaccines are an effort by global leaders to reduce the population size among African nations, that the vaccine might cause infertility, or that vaccinated people might “drop dead” in a few years. Beliefs by some that the COVID-19 vaccine is the biblical “mark of the beast,” with those accepting the “mark” as signifying their allegiance to Satan. Remarks by those vaccinated but still contracting the virus stating: “What’s the point of getting the vaccine if you can still get sick?”. **Challenges**: Transportation to hard-to-reach rural areas and issues of storage and cold chain storage. The extensive use of the internet in the communication of vaccination schedules and other critical information regarding the vaccines meant that older people were left behind. Shortage of doses and their unpredictability impacting programmes. Lack of structured risk communications as well as public health messaging addressing concerns that were exacerbated via social media platforms.
Malawi [180,181,182,183]	**Access**: There were limited quantities of the vaccines available for the population. Various programmes were instigated to increase uptake, including visiting trading places to vaccinate the populous and visiting churches and mobile village clinics. Ongoing programs, in which, if there are at least ten people in a household/homestead/village (or any one place) willing to be vaccinated, they can call authorities who will send a mobile van to vaccinate them. **Acceptance**: Acceptance of COVID-19 vaccines was affected by concerns with their actual effectiveness following misinformation. The other major issue is that the rural people (who are in majority, approximately 80–85% of the population) believe that COVID-19 is a disease for the rich and town/city people, as most recognised deaths are from the educated/rich people who in most cases reside in urban areas. **Hesitancy**: There is high hesitancy among the population, resulting in slow COVID-19 vaccine uptake exacerbated by misinformation that the vaccine will cause infertility, that it is the ‘’mark of the beast (666)’’, as well as the notion that the authorities in Malawi are getting financial rewards for the COVID-19 crisis while the poor should just receive the vaccine. This has resulted in faith leaders being approached to help address misinformation promulgated via social media and other channels. **Challenges**: The vaccination program was faced with limited supply, hesitancy caused by misinformation, and fear of adverse reactions and access challenges. The short lifespan of the vaccine has resulted in the destruction of expired vaccines, which is a concern of wasting valuable resources. Long distances and poor road infrastructures in some areas where people live chalenged attendance at the vaccination sites. This needs to be proactively addressed going forward.
Nigeria [184,185,186,187,188]	**Access**: Initially, access was restricted, with the vaccines only administered in government-owned public facilities. More recently, some state governments (including Lagos State) have included private healthcare facilities. **Acceptance**: There is general apathy and unwillingness to have the vaccine among the general population in Nigeria. This is due to the spread of misinformation and conspiracy theories by those opposed to the COVID-19 vaccines via social media and other platforms. **Hesitancy**: Hesitancy occasioned by the spread of misinformation mainly through social media (WhatsApp, Facebook especially), including misinformation about fertility and adverse effects. There is also the belief that COVID-19 is an illness of the affluent; consequently, there is no need for people in lower socio-economic groups to be vaccinated. The fact that fully vaccinated individuals are infected again has made people believe that there is no need to have the vaccine if one still catches the disease. **Challenges**: Proper storage facilities. Donation of vaccines with short shelf lives (this has led to the destruction of a million vaccine doses recently in Nigeria). Ensuring a steady supply of the vaccines to address disruptions in supply. Addressing the considerable misinformation via social media, as well as enhancing trust in government and other bodies that seek to address pertinent misinformation.
South Africa [90,189,190,191,192]	**Access**: The Electronic Vaccination Data System (EVDS) was introduced with the roll-out for registration of vaccinators, scheduling of appointments and reminders for vaccinees, and generation of personal vaccination records and to support the constant flow of data and information in all directions between area-based, district, and provincial vaccine rollout teams. Department of Health online resource and news portal (https://sacoronavirus.co.za/) 1 August 2022 with a hotline, WhatsApp contact numbers, and email addresses to facilitate communication on all aspects of COVID-19 and vaccination very early in the pandemic. Good public-and private-sector collaboration to open as many vaccination sites as possible across the country. Access was made easier by allowing ‘walk-in’ to clinics with an ID and no need to pre-register and/or make an appointment on the EVDS. Variety of vaccination sites, as well as bringing the vaccines closer to people enhanced access, e.g., mass vaccination sites, pop-up sites, mobile units, drive-through facilities, and vaccination services being available over weekends. Free transport to vaccination sites for people ≥ 50 years old in selected provinces. Incentives created through Vooma vouchers worth R200 for people aged ≥50 years who were vaccinated for the first time (from 1 November 2021 for those ≥ 60 years and from 18 November 2021 for those aged 50–59 years until 28 February 2022). **Acceptance**: Vaccine acceptance reasonable; however, room for improvement. Increase in acceptance was not reflected in the levels of vaccine uptake seen. Inclusion of community organisations was important to ensure vaccination information is addressing community needs and is responsive to social norms. **Hesitancy**: There were concerns with vaccine hesitancy in South Africa influencing uptake rates—with issues of trust in the government being a contributing factor. High levels of hesitancy around the safety of the vaccines. Perceptions of people that their current physical state whether pregnant or breastfeeding, or having an underlying condition, precluded them from getting vaccinated. Other factors influencing hesitancy include age, education, geographical location, employment, and ethnicity. **Challenges**: General low vaccine uptake by the 18–34 years of age cohort. Decrease in vaccine demand from September 2021, while access was not a significant challenge. Vaccination mandates remained a difficult topic in South Africa for most of 2021 Public trust in the government is a concern that needs to be proactively addressed, as it is identified as one of the reasons for the current vaccine hesitancy in South Africa. Low acceptance of vaccines by certain HCWs and influential figures.
Sudan [193,194,195]	**Access**: Vaccines are administered to the target groups at their residential areas in selected PHC facilities to enhance access. Awareness campaigns through different social media and other channels to enhance uptake. **Acceptance**: There is moderate level of acceptance to be vaccinated, although there are variable rates across Sudan and among different groups. **Hesitancy**: Variable levels of hesitancy throughout the country due to circulating misinformation, a particular challenge in South Sudan. There are also high levels of hesitancy among medical students, which needs to be addressed going forward. The source of misinformation is typically social media. **Challenges**: Vertical delivery of the vaccines (with limited use of EPI facilities) makes the distribution sub -optimal. Need to actively address the extent of misinformation that is circulating via social media to reduce current hesitancy.
Tanzania	**Access**: Access is now good with ready availability of the vaccines in all public and some private tertiary health care facilities, overcoming previous concerns. **Acceptance**: Improving, but from low rates of vaccinations initially. Public education has been improved to increase acceptance. **Hesitancy**: Vaccine hesitancy seen in the early days following negative publicity. This is now reducing, leading to higher vaccination rates. **Challenges**: Coverage remains low. Need to continue to address misinformation and beliefs through social media and other platforms to improve vaccination rates from a low base.
Uganda [68,81,196]	**Access**: Very high level; practically all types of vaccines are readily available across the country. **Acceptance**: Overall, fairly good acceptance, although there are variations seen among different groups. **Hesitancy**: Exists especially for Chinese vaccines and for children under 12 years of age, where the school vaccination program widely resisted forcing the ministry to halt the planned roll out. **Challenges**: Expiration of vaccines due to poor uptake in some areas. Inadequate social mobilization.
Zambia [89,131,197]	**Access**: Zambia has 265 designated COVID-19 vaccination sites across the country. Accessibility to the vaccine is higher for those in urban areas compared to those in rural areas. High public sensitisation (through electronic, social, and print media) by the government. **Acceptance**: A large proportion of adult Zambians are currently either uncertain or unwilling to be administered the COVID-19 vaccine, including pharmacy students. A third (33.4%) of surveyed adult Zambians would accept to be vaccinated according to a study that was conducted in April 2021, when the Oxford/AstraZeneca vaccine was just being rolled out. Another survey reported an acceptance of 66% among caregivers in selected districts in Zambia. **Hesitancy**: Vaccine hesitancy from fear of potential adverse effects, concerns about the effectiveness of the vaccines, misinformation regarding the constituents of the vaccines, and barriers to access. Misinformation mostly propagated through social media and religious groups that COVID-19 vaccines cause premature death, including that COVID-19 vaccination is one large experiment and local populations are being used in experiments. Most rural populations perceived COVID-19 as ‘a disease of the West’; consequently, they are uncertain about the need to get vaccinated on account of their belief that COVID-19 poses no risk to them. **Challenges**: Insufficient information provided to the public on vaccines by authorities, especially given the relatively low literacy levels. Strategies to counteract COVID-19 vaccine misinformation are currently not robust enough, with a need for all key stakeholder groups to actively engage in social media activities in the future to address misinformation and its consequences. Inadequate human resource/specialists in immunisation, immunology, and vaccinology to address vaccine-related concerns.
Zimbabwe [93,106,109,175,198]	**Access**: Vaccines are widely accessible both in urban and rural areas throughout Zimbabwe. **Acceptance**: Daily vaccination rates decreased after the third wave, indicating less willingness to get vaccinated, which needs to be addressed going forward. **Hesitancy**: Vaccine hesitancy has resulted in often low demand for the vaccine, especially after the third wave and during the omicron wave where there is low risk perception. Religious beliefs, social media conspiracy theories, lack of trust and confidence in the government, and the desire to have children in the future where the vaccine is perceived to cause sterility are the main reasons for vaccine hesitancy in Zimbabwe. There is widespread misinformation now dubbed “the infodemic” spread through social media, including WhatsApp and Twitter, which can involve HCPs. **Challenges**: Convincing citizens to be vaccinated since vaccinated people still get infections, even though they are less likely to be hospitalized or die. The changing dose regimens with the introduction of booster doses also makes those who are hesitant more sceptical. Need to actively address misinformation promulgated via social media, as well as increase trust in the government and other key stakeholder groups going forward

NB: AEFI = adverse events following immunization; HCPs = healthcare professionals; HCW = healthcare workers; PHCs: Primary healthcare centres.

**Table 5 vaccines-10-01553-t005:** Key activities to improve vaccination rates in current and future pandemics.

Key Activities	Ways Forward
Research activities	Greater involvement of African countries in basic research surrounding the science of infectious diseases and participation in clinical trials is important going forward to help inform the public, building on key activities initially surrounding COVID-19, including genomic sequencing and the local manufacture of ventilators to address shortages [15,174]. This also builds on current research activities among African countries to develop and test vaccines [204]. Improving pharmacovigilance (PV) activities is also crucial for the future to rapidly capture and convey accurate information regarding the effectiveness and safety of current and future vaccines in the real world to address misinformation, especially misinformation promulgated via various social media platforms [120]. This builds on initiatives, including a Pan-African clinical trial registry [205], as well as strengthening current PV activities [206]. Assessing the potential for using performance-based financing and incentives for HCWs, including HCPs, to improve vaccination coverage where there are concerns. Qualitative, context-specific social and behavioural research to complement quantitative surveys to enhance the understanding of perceptions and reasons for vaccine hesitancy in order to help develop future pertinent interventions that build trust and confidence in vaccines and vaccination. Development of context-specific vaccine hesitancy measurement tools to measure the impact of interventions, including those designed for various social media platforms, on vaccine confidence and acceptance and their overall cost-effectiveness, to provide future guidance.
**Education**	
Healthcare professionals (HCPs)	Improving HCPs’ education regarding the science behind vaccines and the importance of the vaccinations for themselves and the general population. This includes addressing key concerns regarding COVID-19 vaccines during university education, as well as continuous professional development post-qualification given current concerns among a number of HCPs. Through educational programmes, community pharmacists, and other HCPs, people can be empowered with additional knowledge and skills to play a key role in addressing misinformation and reducing hesitancy [207,208,209,210]. Such programmes can be part of e-learning approaches, with hybrid learning approaches growing post the current pandemic [10]. Expand on the training of trainers (TOT) approach for capacity development to help address concerns among HCPs, given their increasing role with addressing misinformation and hesitancy and restoring trust [120,175,211,212].
Medical journalists and other key influencers	Governments need to work closely with medical journalists and other key influencers to promote key issues regarding the effectiveness and safety of vaccines to help counter misinformation. This is especially the case for misinformation spread via social media platforms, given their increasing influence and role in promulgating misinformation regarding COVID-19 [120].
General Public	Ensure programmes are in place, including those via pertinent social media platforms, to convey the benefits of vaccines. As a result, enhance uptake as well as address possible areas of misinformation co-ordinated by key national government and regional groups, as well as involve groups such as religious leaders and elders [120,213]. This should also include educational programmes in schools. Social media platforms are considered a key focus area given the rapidity with which misinformation can spread with its subsequent impact on accelerating the inappropriate use of medicines alongside increasing vaccine hesitancy [20,40,120,214,215]. This can also include government educational posters in major sites across countries conveying the benefits of vaccines during the current and future pandemics. Such posters/campaigns need to be in native languages and sensitive to local cultures. Governments need to introduce or enhance online data management systems with key information regarding the effectiveness and safety of vaccines, as well as their availability and uptake, to incentivise people to come forward and be vaccinated. This includes robust systems to capture possible side-effects.
Communication strategies	Context-specific and comprehensive communication strategies for vaccines, aimed at disseminating accurate and transparent information and in a timely manner, are essential going forward. In addition, so is ensuring communication through appropriate social media channels [120,215]. Content of any communication and messaging should be evidence-based, given the level of misinformation seen to date surrounding COVID-19, its prevention with vaccines, and treatment [16,20,28,120]. Collaboration between government, regulatory authorities, civil society, and academia to coordinate the content of communication and speak as one co-ordinated voice. Similar engagement with pharmaceutical companies where there are concerns with the level of misinformation regarding current vaccines and the potential negative impact. Continuous and structured social and community listening to inform communication strategies and interventions to enhance vaccine confidence.
**Engineering**	
Access and availability [216]	Improving access to vaccination, including rural areas, is crucial to future success. This will involve multiple groups delivering the vaccines, including community pharmacists, as well as the use of mobile vaccination clinics [217]. Improving storage facilities for vaccines especially in rural areas, which includes upgrading the available cold chains and the installation and maintenance of solar systems that support cold chains and water supplies. Such activities would help with readily available vaccines during pandemics and minimise the burden on government planning and financing.
**Economics**	
Patient Incentives	Consider offering incentives to the public to enhance vaccination rates. Alongside this, reduce any travelling time to obtain vaccines by ensuring the availability of vaccination sites locally, which can include mobile clinics [217,218].
Local production	Increasing knowledge transfer to enhance the local production of vaccines to help address concerns with possible shortages and to reduce costs and misinformation with current and future pandemics, building on current initiatives [219,220,221,222]. However, there needs to be a market for locally produced vaccines; otherwise, plants may close [223]. In the meantime, African governments should continue with global groups such as the World Bank and GAVI to help secure adequate local supplies of vaccines.
Financial support	Financial support to enable demand creation strategies, social mobilisation, and comprehesive communication strategies incorporating motivational messaging, including pertinent social medial platforms, given their increasing influence [120,224].
**Enforcement**	
Compulsory vaccination [225,226]	Compulsory vaccination programmes were instigated in some countries for key workers, such as those dealing with the elderly, and in at least four countries worldwide for all citizens (Federated States of Micronesia, Indonesia, Tajikistan, and Turkmenistan). However, before instigating compulsory vaccination, there are key legal, ethical, and other issues that need to be considered, including necessity and proportionality, as well as public trust in the effectiveness and safety of the vaccine.
Health system	Integration of COVID-19 vaccination with routine EPI services at the primary healthcare level.

NB: HCPs = healthcare professionals.

## Data Availability

Additional data can be obtained on reasonable request from the corresponding author.

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
