# Peer review of "Coronavirus Disease 2019 (COVID-19) Pandemic across Africa: Current Status of Vaccinations and Implications for the Future"

_vaccines, 2022, doi:10.3390/vaccines10091553_

Round 1

Reviewer 1 Report

Comments and Suggestions for Authors

The paper presents a study aiming to describe the current status of vaccine uptake in Africa, in the domains of access, acceptability, hesitancy, challenges and lessons learnt. In spite of the relevance of this aim, I’m afraid that no clear conclusions are derived from the study. Sadly, because an extensive work has been done, but I doubt if it is worth publishing it as a research paper. Perhaps it could take the form of a report of one of the institutions of the authors, in which its long extension and other considerations could fit better.

I provide comments on several chapters with some concerns, questions, and suggestions.

Introduction

·       The generalities about covid vaccine, medicines against covid, preventive measures adopted in different countries, effectiveness of the vaccines… could be reduced to gain focus on the interesting questions of the study.

·       As a reader, I’d like to have some information on the morbidity and mortality by (or with) covid in Africa, as well as comments on its unregister (Nature paper)

·       The objectives in the abstract are not the same than those in the introduction, neither those in the first paragraph of the results.

·       The authors could add some paragraphs commenting on the amount and accessibility to doses  in these countries

·       The authors could clarify the meanings of acceptance and hesitancy. The data and references about these issues in the introduction interfere with the results of the study.

Methods

·       The narrative review (mainly on the effectiveness and safety of vaccines) seems not relevant for the study. The review of the hesitancy and acceptance of covid vaccines has to be done to set the baseline, the frame and the discussions, but this doesn’t make a mixed methods study, in my opinion. On the other hand, no results of this first review are described in the results. This looks a simply qualitative study gathering information from a small (but perhaps representative) amount of respondents to a guide of topics.

·       Table 1 shows important socioeconomic inequalities among the countries which probably have a great impact on the results. They could have been taken into account, if not analysed, and commented in the discussion.

·       How tand why the respondents were selected?  

Results

·       In general the tables are long. The information in the tables is prolix and heterogenous. It’s difficult to find differences, analogies to make comparisons or find emergent issues. I’d recommend one table with comparable and relevant information, including coverage at one point, for example December 2021, and annex the rest of the tables to supplement information.

·       Coverage would be more understandable if was comparable among countries. Sometimes tables refer to doses administered, to first dose, to completed vaccination, to different moments or periods. Perhaps coverage could be reflected in a figure with the increasing coverage for the participating countries along 2021 from an agency as, for example Our Wordl in Data.

·       An important aspect is lacking: how many doses were available for person (in a period, a year, for example) Which was the flux? How this impacted in the coverages?

·       The indication for children could be presented by date: on June children 5-11

·       The different issues related to hesitancy (patterns, causes, fakes, concrete misinformations, are very interesting!

·       Accessibility was one of the main determinants: availability, logistics, storages, shortatges. Could you describe more deeply them in results?

·       Information systems on the covid vaccines administration were available? In South Africa they look quite solid.

·       The table of key activities reflect the opinions and recommendations of respondents, as they don’t seem being based on the other results? This is not clear.

Discussion

·       Coverages could be discussed with world databases. The study results are not very consistent with Our World in Data, for example.

Author Response

Comments and Suggestions for Authors

  1. a) The paper presents a study aiming to describe the current status of vaccine uptake in Africa, in the domains of access, acceptability, hesitancy, challenges and lessons learnt. In spite of the relevance of this aim, I’m afraid that no clear conclusions are derived from the study. Sadly, because an extensive work has been done, but I doubt if it is worth publishing it as a research paper. Perhaps it could take the form of a report of one of the institutions of the authors, in which its long extension and other considerations could fit better.

Author comments: Thank you for this comment. We have now re-worked the Introduction to concentrate more on key issues including vaccine hesitancy in this continent with the highest prevalence rates of infectious diseases worldwide as well as the highest rate of AMR worldwide. If we do not increase vaccinations rate here (including with COVID-19) – this will have a disastrous impact globally increasing morbidity, mortality and costs.

We beg to differ though regarding your comments on the relevance and conclusions of the paper. We have used senior level personnel from across Africa to provide guidance/ input on the current situation and suggested strategies for the future (Old Tables 5 and 6). Old Table 5 (now Table 4) is an accumulation of suggestions going forward by the senior level co-authors in each of the African countries actively involved in these issues – consequently, able to provide objective guidance based on their considerable experience in this area. We have used this approach before in multiple publications in good impact Journals - to provide guidance on both infectious and non-infectious diseases (referenced). Consequently, we are confident in our approach. As a result, we believe this paper will be well cited and used when and if accepted for publication. For instance, our paper documenting the initial response to COVID-19 across Africa already has already over 33,000 views and multiple citations – we expect similar activities here. We hope you now agree with us.  

  1. b) I provide comments on several chapters with some concerns, questions, and suggestions.

Author comments. Thank you for these – we have addressed these where we can and commented when this is difficult and why.

  1. A) Introduction
  2. i) The generalities about covid vaccine, medicines against covid, preventive measures adopted in different countries, effectiveness of the vaccines… could be reduced to gain focus on the interesting questions of the study.

Author comments: Thank you – we have reduced these. However – still kept important comments to provide a basis for the research. We hope this is OK.

  1. ii) As a reader, I’d like to have some information on the morbidity and mortality by (or with) covid in Africa, as well as comments on its unregister (Nature paper).

Author comments: Thank you for this – we have added this information in as well as concerns with counting the number of deaths across Africa. We hope this is now acceptable.

iii) The objectives in the abstract are not the same than those in the introduction, neither those in the first paragraph of the results.

Author comments: Thank you for this – now addressed to ensure consistency.

  1. iv) The authors could add some paragraphs commenting on the amount and accessibility to doses  in these countries

Author comments: Thank you. We have tackled this in the Tables – concentrating the Introduction on the key issues for the paper. Table 4 also contains details of access/ availability. We hope this is now OK.

  1. v) The authors could clarify the meanings of acceptance and hesitancy. The data and references about these issues in the introduction interfere with the results of the study.

Author comments. Thank you for this – we have addressed this in the revised paper

  1. B) Methods

  1. i) The narrative review (mainly on the effectiveness and safety of vaccines) seems not relevant for the study. The review of the hesitancy and acceptance of covid vaccines has to be done to set the baseline, the frame and the discussions, but this doesn’t make a mixed methods study, in my opinion.

Author comments: Thank you for this. We beg to differ as there should be no/ little hesitancy based on the published effectiveness and safety data with these vaccines. However – the misinformation on social media, etc., is fuelling this along with mistrust in the Government, etc., leading to a high number of avoidable deaths (now added in). We have made this clearer, and hope this is now OK.

  1. ii) On the other hand, no results of this first review are described in the results. This looks a simply qualitative study gathering information from a small (but perhaps representative) amount of respondents to a guide of topics.

Author comments: Thank you. We have now moved old Tables 2 and 3 to the Appendix to concentrate on the main issues of the paper. In this way, readers are not distracted whilst at the same time giving historic data for those who are interested. In addition, used recognised sources to document vaccine rates as well as senior level co-authors to describe key issues and ways forward, building on similar approaches for other situations. As a result, the information gathered, together with the deliberations of the co-authors, has been used to guide future activities especially around issues of social media and government trust. We hope this is acceptable

iii) Table 1 shows important socioeconomic inequalities among the countries which probably have a great impact on the results. They could have been taken into account, if not analysed, and commented in the discussion.

Author comments. Thank you for this – this table is shown to endorse the fact that we have included a wide range of African countries. We have now added in more statistics regarding COVID-19 infection data to the Table to add more depth. We hope this is now OK. However – concerns with hesitancy, etc., appear similar across Africa – hence why we have consolidated the findings when discussing ways forward for across Africa and wider. We hope this is now ok.

  1. iv) How and why the respondents were selected?  

Author comments: Thank you – we included this data in the Methodology but now made this clearer to state that these are typically senior level personnel in their countries (a method that we have used extensively in previous publications to good effect). We hope this is now acceptable

  1. C) Results

  1. i) In general the tables are long. The information in the tables is prolix and heterogenous. It’s difficult to find differences, analogies to make comparisons or find emergent issues. I’d recommend one table with comparable and relevant information, including coverage at one point, for example December 2021, and annex the rest of the tables to supplement information.

Author comments: Thank you – we have now moved some Tables to the Appendix and concentrated on the most recent data as the most relevant especially as supplies of the vaccine have improved – with hesitancy rates, etc. Hope now this is OK.

iii) Coverage would be more understandable if was comparable among countries. Sometimes tables refer to doses administered, to first dose, to completed vaccination, to different moments or periods. Perhaps coverage could be reflected in a figure with the increasing coverage for the participating countries along 2021 from an agency as, for example Our Wordl in Data.

Author comments. Thank you for this. We have sought to clean up the Tables – move some to the Appendix (Supplementary material) and shortened new Tables 2 and 3. This though depends on available data. Thank you for your comments on Our World Data. We had included this as a reference along with WHO data for Africa up to the end of June 2022. These data sources have been consolidated for consistency especially for new Tables 2 and 3. We hope this is now OK.

  1. iv) An important aspect is lacking: how many doses were available for person (in a period, a year, for example) Which was the flux? How this impacted in the coverages?
  • The indication for children could be presented by date: on June children 5-11
  • The different issues related to hesitancy (patterns, causes, fakes, concrete misinformations, are very interesting!
  • Accessibility was one of the main determinants: availability, logistics, storages, shortatges. Could you describe more deeply them in results?
  • Information systems on the covid vaccines administration were available? In South Africa they look quite solid.

Author comments: Thank you for this. As mentioned, we have now concentrated in the main aspects of the paper – which is surrounding issues of vaccine hesitancy and ways forward across Africa, We have though added in further details of uptake rates, etc., at the end of 2021 and middle of 2022 from the same international sources (recommended and WHO) in new/ revised tables as well as added in some comments of storage, etc. in Table 4. We have not gone further that this because - as mentioned - the main aim of the paper is regarding vaccination rates, hesitancy, rationale and ways forward in this very important continent. We hope this is acceptable.

  1. v) The table of key activities reflect the opinions and recommendations of respondents, as they don’t seem being based on the other results? This is not clear.

Author comments: Thank you for this. You are correct – these are based on the experiences, etc., of the senior level co-authors. We have used this technique in multiple publications before involving senior level people across both infectious and non-infectious diseases to good effect including e.g. ways to reduce inappropriate antibiotic use/ AMR to give good guidance to the future, and have made this clear. We trust this is acceptable here.

  1. D) Discussion
  • Coverages could be discussed with world databases. The study results are not very consistent with Our World in Data, for example.

Author comment. Thank you for this. We have enhanced the consistency of these Tables in the Results and Discussion sections, and hope this is now OK.

Reviewer 2

English language and style

( ) Extensive editing of English language and style required
( ) Moderate English changes required
(x) English language and style are fine/minor spell check required
( ) I don't feel qualified to judge about the English language and style

Author comment. Thank you – we have now been through the paper and improved the English with the help of one of the co-authors who is a native English speak with over 450 publications in peer reviewed papers. We hope this is now OK.

Yes

Can be improved

Must be improved

Not applicable

Does the introduction provide sufficient background and include all relevant references?

(x)

( )

( )

( )

Are all the cited references relevant to the research?

(x)

( )

( )

( )

Is the research design appropriate?

(x)

( )

( )

( )

Are the methods adequately described?

(x)

( )

( )

( )

Are the results clearly presented?

(x)

( )

( )

( )

Are the conclusions supported by the results?

(x)

( )

( )

( )

Comments and Suggestions for Authors

This study report the current status of vaccine uptake in Africa. Study findings are important to understand the extant challenges and lessons learnt across Africa to enable the future implementation of such vaccination programmes.

Author comments: Thank you for this summary and kind comments – appreciated.

I have a few observations and suggestions below.

  1. report the vaccine coverage for Tanzania in the abstract, as reported for Egypt.

Author comments: Thank you. Following comments from Reviewer 1 – we have now used 2 main International Sources (Our World and WHO) for consistency to make sure similar remarks in the abstract, main body of the paper and in the Tables. We hope this is now clear.

  1. Below line is hard to understand; consider rewriting.

"from very low in Tanzania up to 53.5% in Egypt at the end 2021, with rates increasing to 87.9% in Egypt by end June 2022."

Author comments: Thank you – now updated.

  1. Please describe in the Methods how the vaccine coverage was ascertained. I understand that the "2.2 Questionnaire design and analysis" section, question 3rd, asks about the coverage from participants, but will this be a credible source of ascertaining coverage?

Author comments: Thank you. As mentioned, we have now used 2 main international sources for coverage (in addition to asking the question) – in this way ensured consistency. We hope this is now OK.

Reviewer 2 Report

This study report the current status of vaccine uptake in Africa. Study findings are important to understand the extant challenges and lessons learnt across Africa to enable the future implementation of such vaccination programmes.

I have a few observations and suggestions below.

1. report the vaccine coverage for Tanzania in the abstract, as reported for Egypt.

2. Below line is hard to understand; consider rewriting.

"from very low in Tanzania up to 53.5% in Egypt at the end 2021, with rates increasing to 87.9% in Egypt by end June 2022."

3. Please describe in the Methods how the vaccine coverage was ascertained. I understand that the "2.2 Questionnaire design and analysis" section, question 3rd, asks about the coverage from participants, but will this be a credible source of ascertaining coverage?

Author Response

This study report the current status of vaccine uptake in Africa. Study findings are important to understand the extant challenges and lessons learnt across Africa to enable the future implementation of such vaccination programmes.

Author comments: Thank you for this summary and kind comments – appreciated.

I have a few observations and suggestions below.

  1. report the vaccine coverage for Tanzania in the abstract, as reported for Egypt.

Author comments: Thank you. Following comments from Reviewer 1 – we have now used 2 main International Sources (Our World and WHO) for consistency to make sure similar remarks in the abstract, main body of the paper and in the Tables. We hope this is now clear.

  1. Below line is hard to understand; consider rewriting.

"from very low in Tanzania up to 53.5% in Egypt at the end 2021, with rates increasing to 87.9% in Egypt by end June 2022."

Author comments: Thank you – now updated.

  1. Please describe in the Methods how the vaccine coverage was ascertained. I understand that the "2.2 Questionnaire design and analysis" section, question 3rd, asks about the coverage from participants, but will this be a credible source of ascertaining coverage?

Author comments: Thank you. As mentioned, we have now used 2 main international sources for coverage (in addition to asking the question) – in this way ensured consistency. We hope this is now OK.

Round 2

Reviewer 1 Report

Thank you for your review